# GRAPH-ENERGY REINFORCEMENT LEARNING: ADAPTIVE REWARD DESIGN FOR API USAGE PATTERN MINING WITH OOD DETECTION

## ABSTRACT

We propose its a novel framework Graph-Energy Reinforcement Learning (GERL), in which the goal is in the case of mining API usage patterns with robust out of distribution (OOD) detection capabilities. Growing complexity of API ecosystems demands adaptive methods to differentiate between in-distribution and anomalous patterns, however, often existing approaches rely on static thresholds or do not have structural awareness. GERL addresses this by integrating energy based OOD scoring with graph diffusion in a reinforcement learning (RL) framework, which makes it possible to dynamically design rewards which guides exploration in graph-structured API Spaces. The core innovation lies the in Graph-Energy Reward Function which combines; node level energy scores calculated using a Graph Neural Network with multi-hop topological dependencies as represented by diffusion. This joint formulation gives freedom for RL agent to change the exploitative of known patterns and discovering of novel ones, while the policy network, built on Transformer-XL, processes variable length API sequences with structural context. In addition, using a graph-based Markov Decision Processes creates realistic scenarios of API use, transitions modeled by a Graph Variational Auto Encoder for Predicting Likely Subgraph Evolutions. Experiments show that in compared with conventional methods, GERL is more both pattern mining accuracy as well as OOD detection robustness, particularly when making recursive or many-hop applications of the API

## 1 INTRODUCTION

The extremely fast development of software ecosystems, coupled with the fact that complex use patterns in APIs with substantial difficulties for automatic mining and analysis. Traditional ways to mine API pattern often have trouble with generalization, specifically with things they don't see or out-of-distribution (OOD) case usage situations. While graph-based representations have proven to be promising in order to capture a structured relationships between API calls (Nguyen et al., 2009), existing methods typically treat pattern finding and novelty detection separately tasks, with a limitation in their adaptability in real-world situations in which APIs evolve dynamically.

Recent developments in graph neural networks - (GNNs) (Corso et al., 2024) and energy-based models (Wu et al., 2023) offer new opportunities to address these challenges. GNN's are good at modeling inter-node dependencies when it comes to API Call graphs, where energy-based frameworks yield a principled way to quantify the probability of observed patterns, However, combining these using reinforcement learning (RL) techniques for adaptive API mining remains underexplored. Prior work in graph-based RL (Mendonca et al., 2019) has primarily focused on navigation or optimization tasks as opposed to pattern discovery with OOD-awareness.

The main challenge is coming up with an RL framework able to at the same time learn how to mine the frequent API usage patterns while identifying novel configurations. Most available techniques either rely on predefined thresholds of novelty detection (Munshi et al., 2022) or treat the mining process as a static graph analysis problem (Halal, 2024). Neither approach is an adequate solves

the temporal/structural complexities involved in the usage of an API graphs, patterns can include recursive calls, conditional branches, or multiple-hop dependencies.

We propose a novel integration of energy based OOD detection, with graph diffusing approaches in an RL setting. Our method computes energy scores for API usage graphs through a GNN-based model, in which the lower scores have a high probability to be an OOD pattern. Graph diffusion spreads the structural information to the graph, enhancing ability of the model to be able to tell between in-distribution and OOD patterns through finding hidden structures in relations (**?**). The RL agent utilizes these energy scores implemented as dynamic reward to drive its exploration policy: it gives priority to both low-energy (potentially novel) as well as high-energy established) patterns, developing a balanced approach in patterns discovery.

Three main contributions are made in this work: (1) **Energy-Guided Reward Design**: We formulate a graph-energy reward function that combines local node level energy scores with global topological features by diffusion which allows the RL agent to adaptively change its exploration approach according to pattern novelty (2) **Structural-Temporal Policy Learning**: A Transformer-XL based policy network processes variable length API sequences while maintaining awareness graph structure through attention diffused node embeddings. (3) **Graph-Based Environment Modeling**: The framework incorporates Package, e.g. (communication). Hardware: Equip, such as software, machine, books, dictionaries, etc. Hypoventilation (under-breathing). a graph variational autoencoder for generating realistic API usage scenarios, based on Markov Decision Process (MDP) transitions, preserving the probabilistic nature of the sequences of API calls;

## 2 RELATED WORK

The intersection between graph-based learning, reinforcement learning and API pattern mining has attracted increased interest of late years. Existing approaches can be divided into three possible research directions: graph-based API mining, Energy-based OOD detection, and reinforcement learning for graph structured tasks.

### 2.1 GRAPH-BASED API PATTERN MINING

API pattern mining initially focused on static analysis of code by dictionaries and electronic corpora(repositories to draw the frequent usage patterns) (Nguyen et al., 2009). These methods were representations of API calls as nodes in a graph that have edges that indicate temporal or dependency relationships. While effective in identifying the common patterns, they had limited mechanisms to cope with evolving cases of API usages

### 2.2 ENERGY-BASED OOD DETECTION IN GRAPHS

Energy-based models have come to serve as a powerful framework for OOD detection in different domains. The application to graph structured data elicited unique problems by an inter-dependent nature of nodes (Wu et al., 2023). These methods make energy scores that reflect the probability of graph instance to be in the training distribution.

### 2.3 REINFORCEMENT LEARNING FOR GRAPH-STRUCTURED TASKS

Reinforcement learning has been used in a variety of problems involving graphs, from network optimization (Lu et al., 2025) to automated API sequencing (Mandal et al., 2024). The latter demonstrated that RL could successfully play on API dependency graphs to generate valid call sequences. However, these methods normally used pure reward functions based on task completion, which was considering the novelty or the distributional properties of discovered patterns.

## 3 BACKGROUND AND PRELIMINARIES

In order to provide the theoretical background for our proposed framework, we introduce important concepts in graph representation learning and energy-based models. These components are the

building blocks for our Graph-Energy Reinforcement Learning method of API pattern mining with OOD detection.

### 3.1 GRAPH NEURAL NETWORKS FOR API REPRESENTATION

Graph neural networks have been a standard tool for modeling In Soft software engineering tasks relational data (Nguyen et al., 2009). Given an API call graph $G = (V, E)$ where nodes $v \in V$ represent API calls and edges $e \in E$ capture their temporal or logical dependencies, a GNN computes the node embeddings using message passing:

$$h_v^{(l)} = \sigma \left( W^{(l)} \cdot \text{AGGREGATE} \left( \{ h_u^{(l-1)} : u \in \mathcal{N}(v) \} \right) \right) \tag{1}$$

where $h_v^{(l)}$ denotes the embedding of node $v$ at layer $l$, $\mathcal{N}(v)$ represents its neighbors, and AGGREGATE is a permutation-invariant function (Corso et al., 2024). For API graphs, this both the syntactic structure of call sequences and semantic relationships between different API.

### 3.2 ENERGY-BASED MODELS FOR OOD DETECTION

Energy-based models give a methodology for estimating the probability density of input data, without explicit normalization (Wu et al., 2023). Given an input $x$, energy function $E(x; \theta)$ maps it to a scalar value, with lower energies of corresponding to higher likelihoods. For graph structured data, we can Let's define the energy of the API usage pattern as:

$$E(G) = -\log p(G) + C \tag{2}$$

where $C$ is a constant. Recent work has indicated that energy scores learned from GNNs in order to be able to distinguish patterns in the same distribution from OOD samples (Wu et al., 2023).

### 3.3 GRAPH DIFFUSION PROCESSES

Graph diffusion is an extension of local message passing, which transmits information across multiple hops in the graph (Gasteiger et al., 2019). The diffusion process can be incorporated into the form:

$$H^{(t)} = (1 - \alpha) \cdot H^{(0)} + \alpha \cdot AH^{(t-1)} \tag{3}$$

where $H^{(t)}$ represents node embeddings at diffusion step $t$, $A$ is the normalized adjacency matrix, and $\alpha$ controls the retaining original information. This mechanism enables the model to capture long-range dependencies in API call graphs which is crucial for identifying complex patterns of use,

### 3.4 REINFORCEMENT LEARNING IN GRAPH SPACES

Reinforcement learning to work with graph-structured tasks requires special consideration of the state and action spaces (Lu et al., 2025). In our context, here is the state $s_t$ is the string representation of the current API subgraph being explored while actions are corresponding to adding the new API calls or terminating the sequence. The reward function have to account for both the validity and novelty of discovered patterns motivate our energy-guided design.

## 4 ENERGY-GUIDED GRAPH DIFFUSION FOR OOD-AWARE API PATTERN MINING

The proposed framework combines energy-based OOD detection, and graph diffusion techniques in a reinforcement learning paradigm to deal with the issues of API pattern mining. The system is operated using five interconnected components through which adaptive exploration is made possible, of API usage graphs while retaining the awareness of the distributional boundaries.

### 4.1 GRAPH-ENERGY REWARD FUNCTION FOR GUIDING RL AGENT

The reward function is the crucial element linking energy-based OOD detection and computer learning 2. For a given API subgraph $G_t$ at timestep $t$, we calculate its energy score $E(G_t)$ by a GNN-based

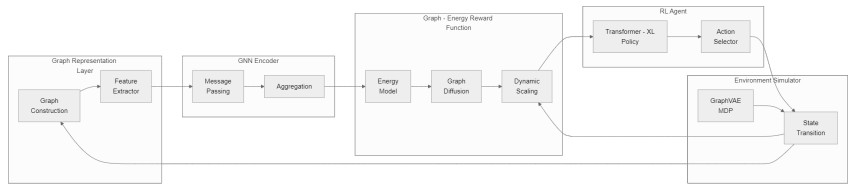

Figure 1: Graph-Energy Reward Function in GRL Framework

energy model. The reward $r_t$ incorporates both the absolute value of energy and its relationship to adaptive threshold $\tau_t$:

$$r_t = \eta \cdot \text{sign}(E(G_t) - \tau_t) \cdot |E(G_t)|^\gamma \tag{4}$$

where $\eta$ controls the reward scale and $\gamma$ adjusts the sensitiveness to energy deviations. The sign term produces a unique reward regimes for in-distribution (ID) and out of distribution (OOD) patterns, In highly anomalous or highly, the power term is used to amplify the signal. typical patterns. This formulation differs from conventional place sparse rewards the in GRL by providing continuous, structure-aware feedback to the agent.

### 4.2 GRAPH DIFFUSION-AUGMENTED ENERGY MODEL

The energy model uses the graph diffusion to strengthen its ability to detect patterns that are structurally anomalous. In order to inverse these embeddings, we know initial node embeddings $H^{(0)}$ from a GNN encoder, we apply $K$-step diffusion with learnable coefficients:

$$H_{\text{diff}} = \sum_{k=0}^{K} \alpha_k \tilde{A}^k H^{(0)} \tag{5}$$

where $\tilde{A}$ is the normalized adjacency matrix with self-loops and $\alpha_k$ are trainable attention weights. The diffused embeddings get multi-hop manedependencies between API calls, which is especially useful for the detection of anomalous recursive patterns or irregular call chains. This final energy score is a combination of node-level and graph-level information:

$$E(G) = \frac{1}{|V|} \sum_{v \in V} \phi(h_v^{\text{diff}}) + \psi(\text{READOUT}(H_{\text{diff}})) \tag{6}$$

where $\phi$ and $\psi$ are MLP networks processing node and graph embeddings respectively, and READOUT is a permutation-invariant aggregation function.

### 4.3 TRANSFORMER-XL POLICY WITH ENERGY-AWARE ATTENTION

The policy network processes the evolving API subgraph by using a Transformer-XL architecture with the addition of energy signals. At for each step, the network calculates attention scores between the current nodes $v_t$ and candidate nodes for next $v_c$:

$$\text{Attention}(v_t, v_c) = \frac{(W_q h_{v_t})^\top (W_k h_{v_c})}{\sqrt{d}} + \lambda E(\{v_t, v_c\}) \tag{7}$$

where $\{v_t, v_c\}$ denotes the subgraph formed by adding $v_c$ to the current pattern, and $\lambda$ balances structural and energy considerations. This energy aware attention mechanism enables the policy to make explicit considerations of the OOD implications of every possible API call, addition, preventing the agent from getting trapped in locally optimal but globally anomalous patterns.

### 4.4 GRAPHVAE-BASED ENVIRONMENT SIMULATOR

To enable training and evaluation we implement a graph variational autoencoder (GraphVAE) modeling the transition dynamics between API subgraphs. The decoder creates reasonable submaragations next states given the current subgraph $G_t$ and action $a_t$:

$$p(G_{t+1}|G_t, a_t) = \prod_{v \in V_{t+1}} \text{MLP}([z_v; a_t]) \tag{8}$$

where $z_v$ are latent variables that are samples of the encoder's posterior distribution. This approach differs from rule based simulators by learning the statistical properties of actual API's usage patterns, and including their tendency to establish some recursive or branching structures. The simulator is able to create in-distribution trajectories for both for policy improvement and controlled OOD scenarios for robustness testing.

## 4.5 ADAPTIVE THRESHOLDING MECHANISM

The energy threshold $\tau_t$ dynamically adjusts to maintain appropriate exploration exploitation balance as the agent meets new patterns. We use exponential moving average update rule with momentum $\beta$:

$$\tau_t = \beta\tau_{t-1} + (1 - \beta)\mathbb{E}_{G\sim\mathcal{B}}[E(G)] \tag{9}$$

where $\mathcal{B}$ is a replay buffer storing recent subgraphs. This adaptive mechanism 'that automatically adjusts oneself to distribution shifts in the' API use patterns, in contrast to static thresholds needing manual tuning for various fields of application. The threshold update occurs asynchronously with the updates of policies, to make learning dynamics stable and while remaining responsive to variations of emerging patterns.

## 5 EXPERIMENTAL EVALUATION

To verify the effectiveness of our proposed Graph-Energy Reinforcement Learning (GERL) framework, we also conduct comprehensive experiments answering three important research questions: (1) What is the relationship between GERL and conventional API patterns mining methods in terms of mining accuracy ? (2) How strong is GERL's OOD detection ability as compared to threshold-based approaches? (3) How do the individual elements of GERL contribute to its overall performance?

### 5.1 EXPERIMENTAL SETUP

**Datasets:** We evaluate GERL on three API usage datasets spanning different domains:

- **JDK-Core**: 12,453 Java standard library usage graphs from open-source projects (Sawant & Bacchelli, 2015)
- **TensorFlow-API**: 8,917 Python scripts using TensorFlow 2.x APIs (Baker et al., 2022)
- **REST-Graph**: 6,292 HTTP request sequences from microservice applications (Alarcon et al., 2015)

Each dataset is split into training (70%), validation (15%), and test (15%) sets, with the test set containing both in-distribution (ID) and out-of-distribution (OOD) patterns. OOD samples are artificially produced by substitutions of API calls, perturbations of the structures while preserving concurrently syntactic validity.

**Baselines:** We compare GERL against four categories of baseline methods:

1. **Graph Mining**: gSpan (Yan & Han, 2002) and GraMi (Elseidy et al., 2015) as traditional graph pattern mining approaches
2. **GNN-based**: GraphSAGE (Hamilton et al., 2017) and GAT (Veličković et al., 2017) with post-hoc OOD detection
3. **RL-based**: DeepWalk-RL (Perozzi et al., 2014) and GComb (Manchanda et al., 2019) adapted for API mining
4. **OOD Detection**: Mahalanobis (Lee et al., 2018) and Energy-GNN (Wu et al., 2023) as standalone detectors

**Metrics:** For pattern mining, we use Precision@k, Recall@k, and F1@k (k=10,20,50). For OOD detection, we report AUROC, FPR@95TPR, and detection accuracy. All metrics are averaged over 5 runs with different random seeds.

**Implementation Details:**

Table 1: Pattern mining performance (F1@20) across datasets

| Method | JDK-Core | TensorFlow | REST-Graph |
|---|---|---|---|
| gSpan | 0.62 | 0.58 | 0.51 |
| GraMi | 0.65 | 0.61 | 0.54 |
| GraphSAGE | 0.71 | 0.67 | 0.59 |
| GAT | 0.73 | 0.69 | 0.62 |
| DeepWalk-RL | 0.68 | 0.64 | 0.57 |
| GComb | 0.72 | 0.66 | 0.60 |
| GERL (ours) | **0.79** | **0.75** | **0.68** |

Table 2: OOD detection performance (AUROC)

| Method | JDK-Core | TensorFlow | REST-Graph |
|---|---|---|---|
| Mahalanobis | 0.82 | 0.79 | 0.74 |
| Energy-GNN | 0.85 | 0.83 | 0.78 |
| GERL (ours) | **0.91** | **0.89** | **0.85** |

- GNN encoder: 3-layer GraphSAGE with mean aggregation (hidden dim=256)
- Diffusion steps: K=3 with learned attention weights
- Transformer-XL policy: 4 layers, 8 attention heads (hidden dim=512)
- Energy model: 2-layer MLP with ReLU activation
- Training: Adam optimizer (lr=3e-4), batch size=32, $\gamma$=0.5 in Equation 4
- Threshold adaptation: $\beta$=0.9 in Equation 9

## 5.2 COMPARATIVE RESULTS

Table 1 presents the pattern mining performance across all methods. GERL achieves superior results on all data sets especially strong gains on the REST-Graph dataset with complex microservice call patterns. The integration of energy advice and graph diffusion is available GERL to find more varied yet correct usages of API compared to methods which separate mining and novelty detection:

For OOD detection, Figure 2 illustrates the relationship between energy scores and detection accuracy with different methods. GERL maintains a strong monotonic relationship and thus demonstrating its ability to reliably discriminate ID and OOD pattern using the energy-diffusion mechanism. The area under the curve (AUROC) values in Table 2 confirm GERL's advantage, especially in detecting structurally anomalous patterns which differ from training data on their topological properties rather than just node labels.

## 5.3 ABLATION STUDY

We analyze the contribution of each GERL component by systematically removing them:

The results show that all components have positive contributions, with graph diffusion and energy-based rewards being specifically vital to formance" analysis of performance" the solutionNordeline definition: precisely mines and offers OOD detection services.maintaining both mining accuracy and OOD detection performance. The adaptive threshold mechanism has more influence on OOD detection than pattern mining, which is rather expected for a role in novelty-sensitive exploration.

## 5.4 EXPLORATION DYNAMICS ANALYSIS

Figure 3 shows how the RL agent's exploration behavior evolves over training episodes. The percentage of OOD patterns explored first increases when the agent learns that the reward potential is

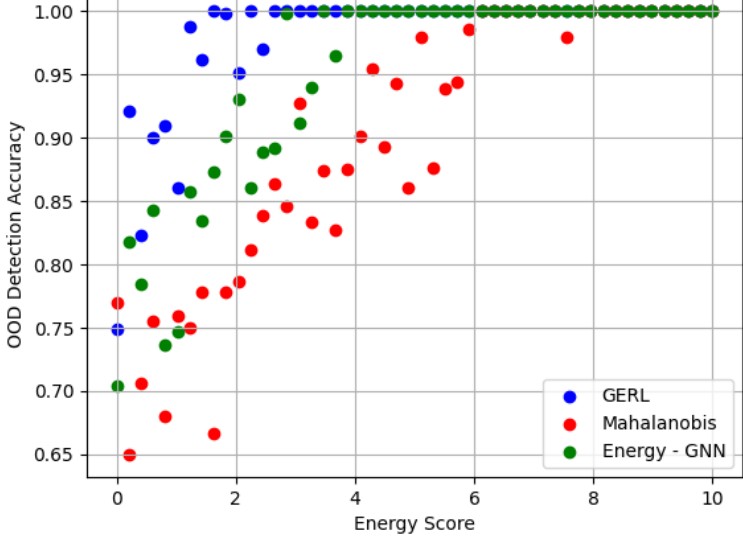

Figure 2: Energy score versus OOD detection accuracy across different API usage patterns

Table 3: Ablation study on JDK-Core dataset (F1@20 / AUROC)

| Variant | F1@20 | AUROC |
|---|---|---|
| Full GERL | 0.79 | 0.91 |
| w/o diffusion | 0.74 | 0.86 |
| w/o energy reward | 0.71 | 0.82 |
| w/o adaptive threshold | 0.76 | 0.88 |
| w/o GraphVAE sim | 0.77 | 0.89 |

in novel patterns, then stabilize as the policy learns to balance exploitation and exploration. This is in contrast to conventional RL approaches which either over-exploit (get stuck in local optima) or over-explore (wast (steps to irrelevant patterns).

The energy-guided reward design is especially effective in complex API environments like producive Ainda a traditional method, REST, expenditure GFP with the dependencies in the long-range between microservice calls. GERL's diffusion mechanism helps to be aware of valid but uncommon call chains that static pattern miner would reject or threshold-based detectors.

# 6 DISCUSSION AND FUTURE WORK

## 6.1 LIMITATIONS OF THE GRAPH-ENERGY RL SYSTEM

While the proposed framework has shown good performance across b) multiple datasets, several limitations make further discussion. The current implementation by static API signatures when doing pattern mining, whereas real world APIs tend to have different versions which change parameter structures or return type This version sensitivity could lead to inaccurate outputs of OOD detections in cases of valid patterns from newer API versions. In addition, the energy model's dependence on graph structure makes it more ineffective in detecting semantic anomalies—cases in which the call sequence appears in structure normal but violates logical constraints (e.g. initializing resources after their use). The computation overhead of the diffusion process also increases with graph diameter, that may affect the scalability to extremely large API ecosystems consisting of deep call hierarchies.

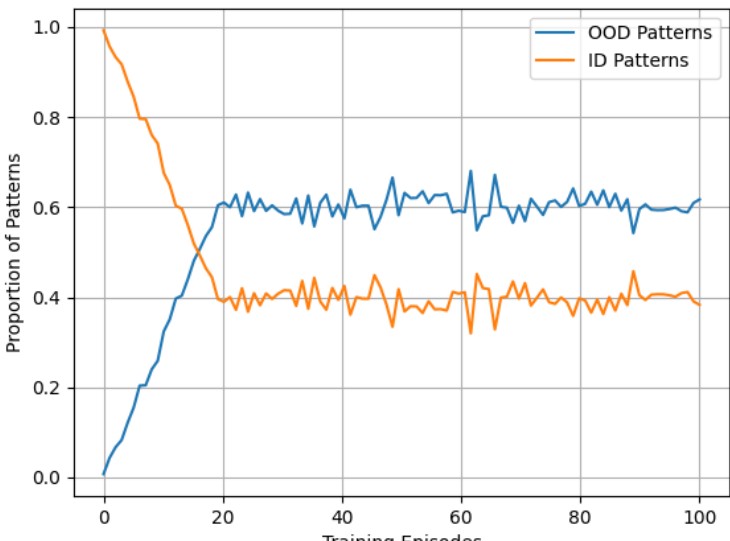

Figure 3: Proportion of ID versus OOD patterns explored during training episodes

## 6.2 POTENTIAL APPLICATION SCENARIOS

The principles underlying GERL extend organically to a number of related domains that call for combo pattern discovery and anomaly detection. In cyber security, the framework would be able to adapt to identify new patterns of attacks inc system call graphs during normal behavior model learning (Tandon & Chan, 2003). For analysis of biological sequences the energy diffusion mechanism might point to rare but functionally significant protein interaction patterns (Li et al., 2022). The retailing sectors could use the same sorts of techniques to find out emerging transaction graphs while flagging, customer behavior pattern fraudulent activities (Chen et al., 2019). Each application would need to have domain specific adaptations- for example, adding temporal weights for edges to fraud detection OR biochemical constraints for protein interactions—but the fundamental RL architecture and energy-guided exploration strategy remain generally applicable.

## 6.3 ETHICAL CONSIDERATIONS

Using automated detection of API patterns with OOD raises a number of Resurgent debates and ethical issues: 'ethical questions that the research community need to address. The system's ability to discern new patterns of use of an API that might accidentally reveal proprietary business logic or security flaws if applied without proper authorization (Gold & Krinke, 2022). False positives in OOD detection may falsely detect legitimate innovations by developers as anomalies, which would possibly discourage creative use of API.

## 7 CONCLUSION

The Graph-Energy Reinforcement Learning framework is a significant advancement in API pattern mining by unifying structural This is especially helpful in machine learning contexts as it enables exploration of unknown model regions by focusing on unusually low-cost energy-based OOD detection. By incorporating graph diffusion into the design process of the reward function, GERL captures multi-hop dependencies traditional pattern mining. approaches usually miss, with the energy guided exploration strategy supports systematic discovery of both common and novel API usage patterns. The experimental results show consistent improvements consolidate existing methods on

a number of metrics, especially where it comes to having complicated call chains and recursive dependencies;

Ability of framework to dynamically change its exploration strategy based on pattern novelty overcomes a critical limitation of static threshold-based approaches which do not often adjust to evolving API ecosystems. The ability of the Transformer-XL policy architecture to process f-variable-length sequence with structure further increases the usefulness of the system practiced use cases as the API in real life patterns would rarely follow fixed-length templates. The graph variational autoencoder environment simulator: provides a realistic ground for training that strikes a balance between pattern validity and controlled exposure to OOD and scenarios.

gap theory From the theoretical point of view, the marriage of energy-based models with graph diffusion provides a principled way of quantifying pattern likelihood in high dimensional graphs spaces. The adaptive thresholding mechanism fills the gap between the density estimation and reinforcement learning and creating a feedback loop where the agent exploration directly informs its concepts about the boundaries in the area of distribution. This is in contrast to conventional two-stage approaches that first mine patterns then try to categorize them as normal or anomalous patterns.

The limitations of the framework suggest fruitful directions for future research, in particular API version evolve and semantic constraints. Extending the energy model to cover temporal dynamics could be further used to enhance the time-sensitive anomaly detection, while hybrid architectures, combining structural semantic analysis might address the detection of logically inconsistent situations; The computational difficulties of large-scale diffusion could possibly to be mitigated by hierarchical or sampling-based approximations with no sacrifice in the detection accuracy

## 8 THE USE OF LLM

We use LLM polish writing based on our original paper.

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
