# OpenReview forum: "Graph-Energy Reinforcement Learning: Adaptive Reward Design for API Usage Pattern Mining with OOD Detection"
_ICLR.cc/2026/Conference — Submitted to ICLR 2026_

### Official Review · Reviewer_NPAm · 2025-10-16

**Soundness:** 2
**Presentation:** 2
**Contribution:** 3
**Rating:** 4
**Confidence:** 3

**Summary:**

The paper proposes Graph-Energy Reinforcement Learning (GERL) for mining API usage patterns with out-of-distribution (OOD) awareness. GERL combines (i) a GNN-derived energy score used to shape a graph-energy reward, (ii) graph diffusion to propagate multi-hop structure into the energy model, (iii) a Transformer-XL policy with energy-aware attention, (iv) a GraphVAE environment simulator, and (v) adaptive energy thresholding. Experiments on three API datasets report gains in pattern-mining F1@20 and OOD AUROC over graph-mining, GNN, RL, and OOD baselines, with ablations indicating contributions from diffusion, the energy reward, thresholding, and the simulator.

**Strengths:**

s1. API usage mining with OOD detection is important for evolving ecosystems.

s2. Unified framework coupling energy signals with diffusion and RL may encourage balanced exploration (novel vs. established patterns).

s3, Reported gains across multiple datasets and metrics, plus ablations indicating component utility.

**Weaknesses:**

w1. GraphVAE transitions are unvalidated. No held-out predictive likelihood, temporal coherence, or structural metric (e.g., motif counts, branching factors) is reported. Without fidelity evidence, policy improvements could overfit simulator artifacts.

w2. Post-hoc OOD on GNN baselines and RL baselines adapted for API mining lack detail. Thresholding strategies, hyperparameter searches, and early-stopping criteria are not provided.

W3. Some strong OOD baselines on graphs are absent.

**Questions:**

Q1. Please clarify whether low energy indicates ID or OOD in your setup.

Q2. Why is sign(E−τ)·|E|^γ preferable to alternatives (e.g., margin losses or asymmetric rewards emphasizing one side)? Did you try advantage shaping or potential-based shaping consistent with policy invariance?

---

### Official Review · Reviewer_mbvj · 2025-10-30

**Soundness:** 2
**Presentation:** 2
**Contribution:** 2
**Rating:** 4
**Confidence:** 3

**Summary:**

The paper introduces Graph-Energy Reinforcement Learning (GERL) to mine API-usage patterns while handling out-of-distribution (OOD) cases. GERL shapes rewards with node-level energy from a GNN and multi-hop graph context and trains a sequence policy over API calls conditioned on graph structure. Across several API graph benchmarks, it reports higher F1@k for pattern mining and better AUROC for OOD detection than baselines, with ablations indicating the energy term and diffusion contribute most.

**Strengths:**

1. Creative integration of energy-guided reward + graph diffusion within an RL policy for API-usage pattern discovery and OOD detection; uncommon crossover of GNNs, energy scoring, and sequence RL.

2. Components are motivated and ablated (energy, diffusion/context, simulator/thresholding). Multi-dataset evaluation with standard metrics (F1@k, AUROC) shows consistent gains over reasonable baselines.

3. Pipeline and training/inference flow are clear; the role of energy in the reward is understandable; problem setting (API sequences on graphs with OOD stress tests) is concrete.

4. Addresses practical needs under distribution shift; if results generalize, likely to influence graph-structured RL/OOD research and tooling for large API ecosystems.

**Weaknesses:**

1. Provide a brief derivation or correlation study showing that higher energy aligns with higher task return.

2. Report stability across seeds, failure cases, and any regularizers used to avoid reward hacking/collapse.

3. Specify the OOD generation procedure, add duplicate/scaffold checks, and include a “hard OOD” split to test robustness.

 4. Add sweeps for diffusion radius/energy weight and a runtime/memory table on larger graphs to demonstrate robustness and practicality.

5. Include at least one stronger recent graph-OOD or energy-guided RL baseline, plus a tuned non-energy RL control with similar compute.

6. Release code and exact split scripts; add a short checklist of hyperparameters/seeds to enable faithful replication.

**Questions:**

1.  Could you spell out the exact definition and whether it’s trained jointly or kept fixed? Any evidence that higher energy actually tracks higher task return across seeds/datasets?

2. Did you see reward hacking or collapse? Please share training curves and variance over ≥5 seeds, and note any entropy/regularization/clipping you used.

3. How sensitive are results to diffusion radius/steps and the energy weight λ? A small sweep with CIs and the reason for your defaults would help.

4. How exactly are OOD samples built and split? What checks avoid near-duplicates or scaffold leakage? A “hard OOD” split would strengthen the claim.

5. A brief runtime/memory table on larger graphs, plus a few details on GraphVAE capacity/validation (to rule out distribution shift), would make practicality clearer.

---

### Official Review · Reviewer_oNwb · 2025-11-01

**Soundness:** 2
**Presentation:** 2
**Contribution:** 2
**Rating:** 2
**Confidence:** 4

**Summary:**

This paper proposes a novel framework called Graph-Energy Reinforcement Learning (GERL) for achieving adaptive out-of-distribution (OOD) detection in API usage pattern mining tasks. The method integrates energy-based models, graph diffusion, and reinforcement learning, enabling the agent to simultaneously discover both common and potentially anomalous patterns while exploring API call graphs.

The main innovations of GERL include:
1.Energy-guided reward design: A reward function based on GNN-derived energy scores allows the agent to dynamically adjust its exploration strategy according to the novelty of the detected patterns.
2.Graph diffusion–enhanced energy model: Multi-hop diffusion propagation is used to capture long-range dependencies among API calls, improving structural awareness and OOD discrimination.
3.Transformer-XL–based policy network: An energy-aware attention mechanism is introduced to handle variable-length API sequences, integrating both structural information and OOD characteristics. The authors further incorporate a GraphVAE environment simulator to model subgraph transition dynamics and design an adaptive energy thresholding mechanism to balance exploration and exploitation.

Experimental results on three datasets—JDK-Core, TensorFlow-API, and REST-Graph—demonstrate that GERL significantly outperforms traditional graph mining, GNN-based, and RL-based baselines in both pattern mining accuracy (F1@20) and OOD detection performance (AUROC). The ablation study further confirms the critical contributions of the graph diffusion and energy-guided reward mechanisms.

**Strengths:**

) Originality
Cross-paradigm integration with novelty:
The paper creatively integrates energy-based models (for OOD scoring), graph diffusion (for multi-hop structural awareness), and reinforcement learning (for policy exploration) into a unified framework, forming an energy-guided reinforcement learning approach for API pattern mining. This cross-domain combination is rarely explored and aligns well with the community’s current interest in OOD robustness.
Mechanistic design:
The authors introduce an energy-guided reward function (Eq.4), a diffusion-enhanced energy representation (Eq.5–6), and a Transformer-XL policy with energy-aware attention (Eq.7), further complemented by a GraphVAE environment simulator and an adaptive threshold updating mechanism. The overall design is systematic rather than a single isolated modification.

2) Quality
Comprehensive metrics and diverse baselines:
Experiments are conducted on three datasets, reporting both pattern mining metrics (F1@k) and OOD detection metrics (AUROC/FPR@95TPR/accuracy). GERL is compared against traditional graph mining, GNN-based, RL-based, and standalone OOD detection methods, showing consistent performance advantages (Table 1 & 2).
Thorough ablation validation:
Removing key components (diffusion, energy reward, adaptive threshold, GraphVAE) leads to noticeable degradation, confirming the necessity of each module (Table 3).
Exploration behavior analysis:
Visualization of exploration dynamics illustrates how energy guidance balances exploration and exploitation, providing useful insight into the mechanism (Figure 3).

3) Clarity
Well-structured presentation:
The method section is clearly organized into submodules (4.1–4.5), with mathematical formulations for all core elements (Eq.4–9), which helps readers understand the end-to-end workflow.
Clear experimental overview:
Dataset sources, splits, evaluation metrics, and implementation details are summarized concisely, lowering the overall replication barrier, though finer details could still be expanded.
Self-reflective discussion:
In the Limitations section, the authors proactively acknowledge version sensitivity, semantic anomaly detection challenges, and diffusion-related computational costs, enabling readers to evaluate applicability boundaries objectively.

4) Significance
Relevance to evolving API ecosystems:
Unifying pattern mining and OOD detection within one framework is practically meaningful, benefiting areas such as software engineering security, quality assurance, and automated maintenance. The paper also highlights potential extensions to cybersecurity, bioinformatics, and fraud detection, underscoring its cross-domain value.
Methodological inspiration:
Introducing energy signals into both RL rewards and attention mechanisms offers a generalizable paradigm for OOD-aware exploration in structured data, potentially inspiring future research in graph representation learning + reinforcement learning.

**Weaknesses:**

1) Inconsistent Energy–OOD Mapping Weakens Reward Design Validity
Issue: The paper suggests in the introduction that “low energy = OOD” to encourage exploration of novel patterns, but later adopts the standard energy formulation “low energy = high likelihood = ID.” These two interpretations contradict each other, directly affecting the sign and scale of the reward function (Eq.4) and undermining its interpretability and correctness.
Evidence: Conflicting descriptions of energy and OOD mapping appear between the introduction and background sections; see the reward formulation in Eq.4.
Suggestion: Unify the definition of the energy–OOD relationship (preferably “low energy = ID, high energy = OOD”), then rewrite the reward’s sign and threshold logic accordingly. Provide stability/convergence analysis and sensitivity experiments for parameters γ,η,λ.

2) Lack of Stability and Convergence Analysis for Adaptive Threshold
Issue: The adaptive threshold employs an EMA update (Eq.9) but lacks analysis of its stability under energy distribution drift, changes in diffusion depth, or graph scale growth. Sensitivity to key hyperparameters β and γ is also missing.
Evidence: The threshold mechanism and related parameters appear in Eq.9.
Suggestion: Add long-term training curves and stability tests under varying β and γ; report reward-scale variance and autocorrelation across training; discuss and prove the fixed-point or convergence conditions under ideal assumptions.

3) Insufficient Description of OOD Construction and Dataset Splitting; Weak Reproducibility and External Validity
Issue: OOD samples are vaguely described as “API replacements or structural perturbations with valid syntax,” lacking details on perturbation strength, ratios, structural constraints, and cross-set leakage control (e.g., similar subgraphs between train/test). Key dataset statistics (graph size, degree distribution, average diameter) are missing, making scalability unclear.
Evidence: Dataset and OOD construction details are briefly mentioned in the experimental setup section.
Suggestion: Publish a full OOD generation protocol and parameter table; include graph structural statistics and complexity scaling curves; evaluate under real version evolution and semantic anomaly cases (syntactically valid but semantically incorrect) to strengthen external validity.

4) Insufficient Baseline Fairness and Statistical Significance
Issue: Although multiple baselines are reported, hyperparameter tuning protocols are undisclosed, and no statistical significance tests or confidence intervals are presented. Only averages are reported, without standard deviation or error metrics, making robustness unverifiable.
Evidence: Results in Tables 1 and 2 and implementation details lack variance reporting.
Suggestion: Provide a unified tuning protocol (search space, early stopping, random seeds); report mean ± standard deviation, confidence intervals, and significance tests (e.g., t-test or non-parametric alternatives); include effect sizes (e.g., Cohen’s d).

5) Insufficient Causal Explanation of Component Interactions (Diffusion, Energy, Threshold, GraphVAE)
Issue: Ablation studies only remove one factor at a time, ignoring interaction effects (e.g., “no diffusion + energy reward,” “diffusion + fixed threshold”). The GraphVAE model error is not quantified, making it unclear how simulation bias affects policy learning.
Evidence: Ablation results appear in Table 3; GraphVAE transition modeling in Eq.8.
Suggestion: Include two-way/multi-factor ablations; report convergence speed, training stability, and policy entropy changes; quantify simulator bias (e.g., KL or EMD divergence from true transitions) and its impact on reward.

6) Presentation and Writing Quality Affect Technical Clarity
Issue: Some sections contain typos, residual edits, and incoherent phrasing, especially in the exploration dynamics paragraph, which hinders comprehension and weakens confidence in experimental validity.
Evidence: Noticeable editing artifacts appear in the exploration dynamics section.
Suggestion: Perform a full language and formatting review; improve figure labeling (legends, axes, units); explicitly describe experimental configurations in Figures 2 and 3 (e.g., threshold curves, energy range).

**Questions:**

1.Unify the Energy–OOD Mapping and Revise the Reward Design
The current text inconsistently defines whether low energy represents ID or OOD, which directly affects the sign and magnitude interpretation of Eq. 4 and thereby the policy learning direction.

The authors should provide:
A clear and unique definition of the energy ↔ distribution property mapping (recommended: low energy = ID, high energy = OOD);
A rewritten or confirmed reward and threshold formulation based on this definition (including the role of sign(E − τ) and ∣E∣^γ;
Evidence of stability, convergence, and sensitivity through ablation or theoretical analysis of parameters γ,η,λ.

2.Fully Disclose OOD Construction and Dataset Splitting Protocol to Prevent Leakage and Strengthen External Validity
The current paper does not specify perturbation strength, ratio, structural preservation constraints, or cross-set subgraph-similarity control, which compromises reproducibility and credibility.

The authors should provide:
A complete OOD generation protocol and parameter table (types, strengths, and ratios of perturbations; node/edge-level structural constraints);
A data-splitting strategy ensuring no leakage (e.g., subgraph deduplication, hashing, or isomorphism detection);
Graph structural statistics (size, degree distribution, diameter) and at least one semantic-anomaly or version-evolution evaluation to verify external validity.

3.Ensure Baseline Fairness and Statistical Rigor
Only mean results are reported—no variance or significance analysis—and baseline tuning procedures are not transparent, making it difficult to confirm that improvements are truly significant.

The authors should provide:
A unified tuning and training protocol (search space, early-stopping criteria, random-seed count, time budget);
Mean ± standard deviation/confidence intervals, significance tests (t-test or non-parametric), and effect sizes (e.g., Cohen’s d);
Multiple random-seed repetitions and error-bar visualizations for key results.

---

### Meta-Review · Area_Chair_qNB3 · 2025-12-28

**Summary:**

This paper proposes a framework named Graph-Energy Reinforcement Learning for mining API usage patterns while also providing OOD detection. The core idea is to couple a GNN-derived energy score for API-usage graphs with graph diffusion to incorporate multi-hop structure, and then use these energy signals to shape rewards for an RL agent that explores graph-structured API call sequences.
The policy is described as a Transformer-XL operating over variable-length sequences with structural/energy-aware attention, and the environment dynamics are modeled via a GraphVAE-based simulator. The paper reports empirical gains on several datasets on both pattern mining and OOD metrics, and includes ablations over diffusion/reward/threshold/simulator components.

**Reviewer Concerns:**

All three reviewers find the direction interesting (energy + diffusion + RL for structured API mining under distribution shift) and acknowledge that the paper evaluates on multiple datasets with ablations. However, the reviewers consistently raise major concerns about technical soundness/clarity and experimental rigor, which collectively prevent confidence in the claims at this stage. In particular, multiple reviewers point to a central inconsistency in the paper’s energy-ID/OOD interpretation, which directly affects the reward design and the validity of the proposed method, along with gaps in OOD protocol specification, baseline fairness, and simulator validation.

**Reviewer Scores:**

The authors did not respond to the reviews (no rebuttal/author discussion was provided). As a result, the reviewers’ major issues/questions, particularly around the energy-ID/OOD definition, OOD generation protocol, training stability, and simulator validation, remain unaddressed in the rebuttal, and there is no additional evidence enabling the AC to resolve these concerns in the authors’ favor.

---

### Decision · Program_Chairs · 2026-01-26

Reject